# AMORE-Isoprene v1.0: A new reduced mechanism for gas-phase isoprene oxidation

Forwood Wiser[1], Bryan Place[2], Siddhartha Sen[3], Havala O.T. Pye[4], Benjamin Yang[5,9], Daniel M. Westervelt[5,6], Daven K. Henze[7], Arlene M. Fiore[8,9], and V. Faye McNeill[1,9]

[1]Department of Chemical Engineering, Columbia University, New York, NY USA 10027
[2]ORISE Fellow at the Office of Research and Development, Environmental Protection Agency, Research Triangle Park, NC, USA 27711
[3]Microsoft Research, New York, NY, USA 10012
[4]Office of Research and Development, Environmental Protection Agency, Research Triangle Park, NC, USA 27711
[5]Lamont-Doherty Earth Observatory of Columbia University, Palisades, NY, USA 10964
[6]NASA Goddard Institute for Space Studies, New York, NY, 10025
[7]Department of Mechanical Engineering, University of Colorado, Boulder, Boulder, CO, USA 80309
[8]Department of Earth, Atmospheric, and Planetary Sciences, Massachusetts Institute of Technology, Cambridge, MA USA 02139
[9]Department of Earth and Environmental Sciences, Columbia University, New York, NY USA 10027

**Correspondence:** V. Faye McNeill (vfm2103@columbia.edu)

**Abstract.** Gas-phase oxidation of isoprene by ozone ($O_3$) and the hydroxyl (OH) and nitrate ($NO_3$) radicals significantly impacts tropospheric oxidant levels and secondary organic aerosol formation. The most comprehensive and up to date chemical mechanism for isoprene oxidation consists of several hundred species and over 800 reactions. Therefore, the computational expense of including the entire mechanism in large-scale atmospheric chemical transport models is usually prohibitive, and most models employ reduced isoprene mechanisms ranging in size from $\sim$10 to $\sim$200 species. We have developed a new reduced isoprene oxidation mechanism using a directed graph path-based automated model reduction approach, with minimal manual adjustment of the output mechanism. The approach takes as inputs a full isoprene oxidation mechanism, the environmental parameter space, and a list of priority species which are protected from elimination during the reduction process. Our reduced mechanism, AMORE-Isoprene, consists of 12 species which are unique to the isoprene mechanism and 22 reactions. We demonstrate its performance in a box model in comparison with experimental data from the literature and other current isoprene oxidation mechanisms. AMORE-Isoprene's performance for predicting the time evolution of isoprene oxidation products, including isoprene epoxydiols (IEPOX) and formaldehyde, is favorable compared to other similarly sized mechanisms. When AMORE-Isoprene is included in the Community Regional Atmospheric Chemistry Multiphase Mechanism 1.0 (CRACMM1AMORE) in CMAQ v5.3.3, $O_3$ and formaldehyde agreement with EPA Air Quality System observations are improved. $O_3$ bias is reduced by 3.4 pbb under daytime conditions for $O_3$ concentrations over 50 ppb. Formaldehyde bias is reduced by 0.26 ppb on average for all formaldehyde measurements compared to the base CRACMM1. There was no significant change in computation time between CRACMM1AMORE and the base CRACMM. AMORE-Isoprene shows a 35% improvement in agreement between simulated IEPOX concentrations and chamber data over the base CRACMM1 mechanism

when compared in the F0AM box model framework. This work demonstrates a new highly reduced isoprene mechanism and shows the potential value of automated model reduction for complex reaction systems.

## 1 Introduction

Isoprene is the most abundant non-methane hydrocarbon in the atmosphere. It has a major impact on tropospheric oxidant levels (Butler et al. (2008)) and contributes to secondary organic aerosol (SOA) formation (and therefore fine particulate matter, $PM_{2.5}$) in many parts of the U.S. and the world (Kroll et al. (2006); Henze and Seinfeld (2006); Farmer et al. (2010); Liu et al. (2016); Fu et al. (2009)). During the warm season, isoprene emissions enhance both regional and hemispheric ozone abundances at northern mid-latitudes (Fiore et al. (2011); Guo et al. (2018)). Isoprene oxidation chemistry contributes to natural background ozone and particulate matter over much of the U.S. during the warm season (Fiore et al. (2014)). Different representations of isoprene chemistry lead to uncertainty in air pollutant responses to anthropogenic emission reductions (Carlton et al. (2009)), and differences in model estimates of the background versus anthropogenic fractions (Fiore et al. (2005); Fiore et al. (2014)).

Knowledge of the isoprene oxidation reaction mechanism, including key pathways for both ozone and aerosol formation, has advanced rapidly over the last two decades (Wennberg et al. (2018)). The full chemical mechanism for isoprene oxidation, as it is currently understood, consists of several hundred species (up to 602 in MCM v3.3.1 (Jenkin et al. (2015))) and ∼1000 reactions (see Table 1). Due to its size and complexity, including every known intermediate species and reaction in the isoprene oxidation network in 3D air quality and atmospheric chemistry models is not feasible. Therefore, most models employ reduced isoprene mechanisms. For reduced mechanisms, there is a trade-off between mechanism size (i.e., the number of species and reactions represented) and accuracy. The goal is to find the smallest possible reduced mechanism that still provides the accuracy required for the modeling application. Commonly used reduced isoprene mechanisms range in size from ∼10 to ∼200 species. Table 1 shows the size of a select set of isoprene mechanisms currently being used in atmospheric chemistry models. The reduced models, including the Common Representative Intermediates mechanism, Caltech Reduced Plus mechanism, Regional Atmospheric Chemistry mechanism, and Carbon Bond mechanism have been developed manually by expert air quality scientists using techniques such as surrogate mechanisms (lumped structure (Yarwood et al. (2005)) or lumped species (Aumont et al. (2005); Goliff et al. (2013); Jenkin et al. (2019))), and empirical parameterization, along with expert knowledge of the reaction system. While these approaches have been successful in representing atmospheric chemistry for the specific chemical and environmental scenarios for which they were developed, the resulting models tend to lack flexibility to be adapted to new scenarios or to be rapidly updated. Their implementation is also labor intensive.

Automated chemical mechanism reduction techniques provide the opportunity to flexibly and rapidly generate accurate reduced chemical mechanisms, and lower the barrier to updating the mechanism as new knowledge becomes available. While automated mechanism reduction has been applied in limited studies in atmospheric chemistry (Whitehouse et al. (2004a); Whitehouse et al. (2004b); Watson et al. (2008); Xia et al. (2009); Nikolaou et al. (2018), Sturm (2021), Kelp et al. (2022), Lin et al.), it has been further developed in the field of combustion (Wei and Kuo (1969); Tomlin et al. (1992); Tomlin et al. (1997);

**Table 1.** Sizes of select isoprene mechanisms including the mechanism in this work. Larger mechanism sizes are self-reported. For smaller mechanisms, species and reactions were recounted for this work using the following criteria: i) only species unique to isoprene chemistry are included, which excludes species that do not contain the isoprene carbon backbone. ii) All reactions involving species unique to isoprene are counted iii) Heterogeneous reactions involving isoprene species are not counted.

| Mechanism | Number of species | Number of reactions | Reference |
| --- | --- | --- | --- |
| MCM 3.3.1 | 602 | 1926 | Jenkin et al. (2015) |
| CRI 2.2 | 56 | 186 | Jenkin et al. (2019) |
| Caltech | 404 | 897 | Wennberg et al. (2018) |
| Caltech Reduced Plus | 131 | 220 | Wennberg et al. (2018) |
| RACM2 | 9 | 12 | Sarwar et al. (2013), Goliff et al. (2013) |
| CB6r3 | 10 | 17 | Yarwood et al. (2010); Emery et al. (2015) |
| AMORE-Isoprene | 12 | 22 | This work |

Massias et al. (1999); Lu et al. (2001); Lu and Law (2005); Pepiot-Desjardins and Pitsch (2008); Sun et al. (2010)). Combustion mechanisms have a number of features in common with the isoprene oxidation mechanism, including their complexity and the large number of intermediates involved. Thus, techniques developed for the application to combustion mechanisms may be
applicable to the isoprene oxidation mechanism as well.

The methods of model reduction, whether automated or manual, fall into two main categories. The first is reduction by removing less important species or reactions. The other method is to group species and reactions together which may participate in similar reaction pathways (chemical lumping). Each method aims to reduce the computational cost for simulating the mechanism by reducing the complexity and size of the reaction network, while retaining accuracy within a given tolerance.
Graph theory has been used as a framework for many model reduction algorithms, including the ones used in this work. An influential method for reduction is the Directed Relation Graph (DRG) method, developed by Lu and Law (2005). In this method, a graph representing the reaction mechanism is created, consisting of nodes (carbon-containing chemical species) connected to each other by directed edges (reactions). Each edge is given a weight based on the strength of the relationship between the two nodes, which is a function of the kinetic rate laws and parameters. In the model reduction process, edges are
removed in order of the weightings. Other methods include variations on the edge weighting calculation to include more indirect relationships between species. One such method, the Directed Relation Graph with Error Propagation (DRGEP), was used to reduce the RACM atmospheric mechanism (Nikolaou et al. (2018)). This method was successful at significantly reducing the number of species while maintaining accuracy for simulating $O_3$. We tested the DRG method and found it to be unsuitable for application to the isoprene mechanism (see section S.3). Briefly, the DRG method is successful when a mechanism features a
significant number of species that can be removed without major consequence to the desired accuracy. The scale of reduction required for the isoprene mechanism, and breadth of important priority species (Section 2.2), make it incompatible with the DRG method.

Here, we present a new reduced isoprene oxidation mechanism that we have developed using a novel graph theory-based Automated Model Reduction approach (AMORE), with minimal manual adjustment of the output mechanism (AMORE-Isoprene). We describe the model reduction algorithm, then demonstrate the performance of AMORE-Isoprene compared to experimental data in the literature and other isoprene oxidation mechanisms using a box model, and when incorporated into the Community Multiscale Air Quality (CMAQ) modeling system (U.S. EPA Office of Research and Development (2021)) as part of the Community Regional Atmospheric Chemistry Multiphase Mechanism (CRACMM1AMORE).

## 2  Methods

In this section we describe our approach for model reduction and inputs to the process, and the procedure used for testing the reduced mechanism.

In brief, an algorithm was developed to reduce the full isoprene mechanism to a smaller more manageable mechanism that can be used in 3D chemical transport models. The output mechanism from the AMORE algorithm was subsequently adjusted manually to optimize its performance for use in atmospheric modeling. In order to test the AMORE-Isoprene mechanism, a mechanism error metric was devised.

The AMORE-Isoprene mechanism was the product of this methodology. Our novel algorithm was essential in the creation of this mechanism, but requires further work before it can be used for other mechanisms and without manual adjustment.

### 2.1  Full Mechanism Input

A "full" chemical mechanism is required for the input to the reduction algorithm. The full mechanism also serves as a benchmark for the accuracy of the reduced mechanism. In this study, the reference isoprene oxidation mechanism was based on Wennberg et al. (2018). The Wennberg mechanism is a comprehensive compilation of isoprene oxidation chemistry from laboratory and computational studies published up to 2018, including the formation of isoprene epoxydiols (IEPOX) (Paulot et al. (2009)), intramolecular $RO_2$ chemistry (autoxidation) (Teng et al. (2017)), and recent advances in isoprene nitrate chemistry (Schwantes et al. (2015)). Despite its size and complexity, some branches of the oxidation cascade are truncated in the Wennberg et al. (2018) mechanism due to lack of published experimental constraints, specifically degradation pathways for some later-generation intermediates with 2, 3, or 4 functional groups (Bates and Jacob (2019)). Therefore, modeled on the approach used by Wennberg et al. (2018) in preparing the Caltech Reduced Plus mechanism, we expanded the Wennberg mechanism to include degradation of these species. Further details are available in section S.1 including box model comparisons of original and extended mechanisms to EUROCHAMP data (Muñoz (2021a), Muñoz (2021b)). In addition, the extended mechanism is listed in its entirety in section S.19. Briefly, the intermediates were mapped to lumped species in the Caltech Reduced Plus mechanism or species in MCMv.3.3.1, and assigned the corresponding degradation pathway, products and rate constants from that mechanism. For the rest of this manuscript, we refer to this updated mechanism as the *Caltech full mechanism*. This mechanism was chosen instead of the MCM isoprene mechanism (Jenkin et al. (2015)), which is of a similar size,

because it includes the results of more recent isoprene chamber studies which were not yet published at the time that the current MCM mechanism was developed (e.g., Teng et al. (2017)).

## 2.2 Priority Species

Given that model reduction necessarily involves removing or lumping chemical species from the mechanism, we identified a set of nine important organic species and eight important oxidant and nitrogen oxide species to be protected from elimination during the model reduction process. This priority species list was an input to the model reduction algorithm. A full table of these species is available in section S.2. Besides isoprene, these species were chosen for their importance for SOA or brown carbon formation and/or expected impact on gas-phase photochemistry (isoprene epoxydiols (lumped), isoprene nitrates (lumped), glyoxal, methylglyoxal, methacrolein, methyl vinyl ketone, peroxyacetyl nitrate, methyl radical, peroxyacetyl radical). Formaldehyde was also included in the protected species list due to its status as an air toxic (EPA (2018); Zhu et al. (2017); Scheffe et al. (2016)) and for its potential to indicate oxidant levels (Travis et al. (2022)). Other species such as $NO_x$, $HO_x$, $O_3$, and other oxidants are included in the mechanism as well. The accuracy of the reduced isoprene mechanism is measured by its ability to simulate the time evolution of the concentrations of the priority species and oxidants and nitrogen oxides under different conditions.

## 2.3 Reduction Algorithm Development

In general, a new reduced isoprene oxidation mechanism will be a good candidate for application in large scale models if it provides gains in accuracy or computational efficiency. Since a tradeoff exists between mechanism size (and therefore computational efficiency) and accuracy, improvements in one aspect are sought which avoid sacrifices in the other. Therefore, the mechanism should be of similar size and complexity to existing mechanisms (or smaller), and of equal or better accuracy. The most compact isoprene mechanisms, including those currently used in the CMAQ modeling suite (RACM2 and CB6r3), include roughly 10 species unique to the isoprene mechanism and up to 20 reactions (Table 1). Note that this list of species does not include all priority species; some, such as IEPOX and isoprene nitrates, are included, whereas others, such as formaldehyde and glyoxal, which lack the isoprene carbon backbone and are also formed through non-isoprene pathways, are not. Thus, an isoprene mechanism of comparable size to existing reduced mechanisms will have around 10 isoprene specific species, around four of which (isoprene, isoprene nitrates, IEPOX, methyl vinyl ketone), are already priority species. The remaining 6 species are isoprene intermediates which are not considered priority species themselves, but play an important role in the dynamics of the isoprene mechanism and the production of priority species.

The AMORE algorithm represents the full mechanism as a graph. Many prior works have utilized graph theory to analyze chemical mechanisms (Ratkiewicz and Truong (2003) , Lu and Law (2005), Pepiot-Desjardins and Pitsch (2008), Sun et al. (2010), Nikolaou et al. (2018), Silva et al. (2021)). In this work, nodes represent species, and edges represent a directed relationship between two species, in which one is a reactant and the other a product of the same set of reactions. Prior graph-based reduction methods have focused solely on removing non-essential components of the mechanism ('pruning' the graph). This work focuses instead on determining the optimal graphical structure of the final reduced mechanism, as constrained by

target mechanism size. This is done by determining the essential mechanistic pathways needed to accurately represent the full mechanism in a reduced structure, as discussed below.

A mechanistic pathway consists of a set of reactions joined by intermediate species. For a path of N reactions, there are N - 1 intermediates. With the constraint of 6 intermediate species, this allows for roughly 6 paths with 2 reactions each with one intermediate, or 3 paths with 3 reactions each with two intermediates, both options having 6 intermediates. If some pathways are able to share intermediates, then more pathways can be included. It is also our goal that the reduced mechanism structure maps as closely as possible to known reactions with measured rates.

A new algorithm was designed specifically to develop optimal mechanisms of roughly 10 total species, 6 intermediate species, and 20 reactions. At a high level, the algorithm identifies a small set of the most important mechanistic pathways in the full mechanism, and concatenates them in order to reduce the number of intermediate species. The algorithm estimates the importance of a given mechanistic pathway by determining the impact each possible pathway has on the yields of priority species. The mechanism reduction algorithm has four main components: (1) a sub-algorithm to rapidly estimate the yields from isoprene of priority species under constant oxidant and nitrogen oxide concentrations and atmospheric conditions (yield estimation algorithm, Section 2.3.1), (2) a sub-algorithm to assess the importance of different pathways given the yields of priority species (pathway importance algorithm, Section 2.3.2), (3) a sub-algorithm for optimally combining pathways to reduce intermediate species (pathway combination algorithm, Section 2.3.3), and (4) a sub-algorithm to estimate yields of priority species for each pathway in the mechanism (priority species yield determination, Section 2.3.4). The overall AMORE algorithm process is shown in Figure 1. All sub-algorithms are described in detail in the following sections.

### 2.3.1 Yield Estimation Algorithm

The yield estimation algorithm utilizes graph theory, and takes advantage of the relatively small number of cycles (a path in the graph that starts and ends at the same species) and small number of reactions with two carbon-containing reactants in the isoprene oxidation scheme. It rapidly estimates the yields of all species from isoprene in the full mechanism, assuming the complete oxidation of isoprene and its products. The algorithm emulates the full mechanism so that the numerical simulation need not be run repeatedly during sensitivity testing. The algorithm begins by representing the full mechanism as a directed graph. The directed nature of the representative graph delineates the direction of the flow of carbon over time. Cycles are unique instances in this context, in which carbon flows in two different directions and it is not necessarily evident which direction dominates. The algorithm takes oxidant and nitrogen oxide concentrations (OH, $HO_2$, $O_3$, $MO_2$, NO, $NO_2$, $NO_3$), which are treated as constant, solar intensity, temperature, and pressure as inputs, and calculates the flux of carbon through the mechanism pathways using the rate law information provided. Since this algorithm is dependent on oxidant and nitrogen oxide concentrations and other atmospheric parameters, it can be used to determine how yields are impacted by relevant atmospheric conditions.

The full mechanism is approximated using a directed acyclic graph (DAG). In order for a mechanism to be represented as a DAG, it must contain no cycles and reactions with two reactants must be broken into two sets of edges for each reactant, because edges can only represent the relationship between two species. For example, a reaction with two reactants and one

**1. Yield Estimation Algorithm**

Isoprene $\longrightarrow$ $Y_{HCHO}$ + $Y_{IEPOX}$ + ... = **Y** = priority species yields

$W_{HCHO}$ × $Y_{HCHO}$ + $W_{IEPOX}$ × $Y_{IEPOX}$ + ... = **w·Y**
= weighted priority species yields

**2. Path Importance Algorithm**

**Y**{OH, NO} = **Y**(elevated [OH], elevated [NO], else baseline)

I = min ( **w·Y**{OH, NO} - **w·Y**{OH}, **w·Y**{OH, NO} - **w·Y**{NO})

**3. Path Combination Algorithm**

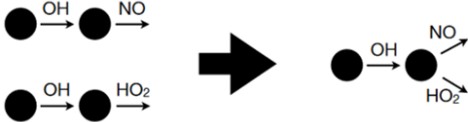

**4. Priority species yield determination**

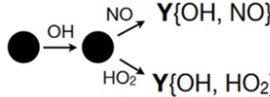

**Figure 1.** Schematic of the AMORE algorithm. The sub-algorithms are shown in order of implementation. Brackets are used specify a pathway within the mechanism, with each oxidant or nitrogen oxide within the brackets representing a reaction in a sequence involving that oxidant. For example, the pathway {OH, NO} represents a sequence of two reactions joined by an intermediate, in which OH and NO are reactants in the two reactions. The pathway {OH, NO} is shown as an example for sub-algorithms 2-4.

product would become two edges, one for each reactant connecting to the product. Oxidant and nitrogen oxide concentrations are approximated to be constant, so reactions involving them are treated as pseudo first-order. Cycling in the isoprene oxidation system mainly takes place among oxidant and nitrogen oxide species, which are only represented implicitly in the graph. For cycles involving isoprene oxidation products, all species in the cycle are combined into one 'super node'. The incoming and
175    outgoing edges of the super node include all edges of all species that it represents. The method used to reduce cycles to super nodes is described in section S.4.1.

     The DAG structure is then utilized to calculate the partitioning of carbon between branches within the graph, ultimately giving an estimated yield for each oxidation product species. This novel approach takes advantage of the graphical representation of the mechanism to rapidly approximate yields which would otherwise require a box model to calculate. The resulting time
180    savings allows a much larger set of input conditions to be tested than would be feasible with a box model. The yield is defined as the moles of each species produced per mole of isoprene reacted. Starting from a species of known yield, the yield of a direct product can be calculated as the rate constant involving said product over the sum of all rate constants reacting with the starting species. The yield for a species A from isoprene, $Y_A$, is calculated as follows:

$$Y_A = \sum_{i=1, i \neq A}^{N} Y_{i,isop} Y_{A,i} \tag{1}$$

$$Y_{A,i} = \frac{\sum_{r=1}^{R} k_r^I max(\nu_{A,r}, 0)(-min(\nu_{i,r}, 0))}{\sum_{r=1}^{R} k_r^I(-min(\nu_{i,r}, 0))} \tag{2}$$

where $Y_i$ is the yield of species i from isoprene, and $Y_{A,i}$ is the yield of species A from species $i$, $N$ is the number of species, $R$ is the number of reactions, $\nu_{i,r}$ is the stoichiometric coefficient of species $i$ in reaction $r$, and $k^I$ is the (pseudo) first order rate constant, that is, $k^I = k^{II}[oxidant]$ for oxidation reactions, or else the first order rate constant for photolysis and isomerization reactions. The yield of any species can be estimated once the yields of all its parent species in the graph are determined. Thus, with the assumptions and inputs outlined above, an estimate of the yield of all species from isoprene can be obtained for a given set of inputs. Running in a Jupyter notebook environment on a personal computer with a 1.8GHz dual-core Intel Core i5 processor, it takes roughly 0.06 seconds for the algorithm to estimate all yields for a given set of conditions, 50x shorter than a box model run time of the full mechanism. This is a valuable tool for rapidly probing large mechanisms to study their outputs under a variety of inputs.

The yield estimation algorithm was tested for accuracy by comparing estimated yields to box model simulated yields for the range of conditions used for model development. A detailed analysis of the yield estimation algorithm accuracy is available in section S.4.2. A visualization of the yield estimation algorithm is shown in Figure S.8.

### 2.3.2 Pathway Importance Algorithm

With the yield estimation algorithm in place, we developed a method to identify and evaluate the importance of paths within the mechanism. Given the constraints on the size of the final reduced mechanism discussed, the total number of paths will vary depending on the number of intermediates that can be shared between paths. The full mechanism contains long, highly branched paths with multiple end products. Thus, no existing pathways within the full mechanism satisfy the design constraints. Instead, model paths were created in which each path was represented by a sequence of reactions with one of the possible oxidants or nitrogen oxides: OH, NO, $NO_2$, $NO_3$, $HO_2$, $O_3$, methyl peroxy radical ($MO_2$), or else photolysis. There was no requirement for a given path to be in the full mechanism, rather paths recreate the oxidant and nitrogen oxide dependent outcomes for the priority oxidation products. Each path was constrained to contain only irreversible reactions, with each oxidant or nitrogen oxide appearing no more than once (this constraint was lifted during the manual adjustment process). The justification for these simple paths is that isoprene oxidation product concentrations can be thought of as functions of isoprene, oxidant, and nitrogen oxide concentrations, and each path represents a scenario in which a set of oxidants and nitrogen oxides are favored. Thus, by containing multiple different paths, the priority species yields can be varied based on the oxidant and nitrogen oxide concentrations. For example, a path of {OH, NO} represents the reaction of OH or NO with isoprene to create a hypothetical intermediate, and the reaction of the other oxidant or nitrogen oxide (either NO or OH) with that intermediate to form isoprene oxidation products. This path would be favored when OH and NO concentrations are high, and allows for a unique distribution

**Table 2.** Baseline and elevated values of input parameters used in the pathway importance algorithm, a component of the AMORE algorithm.

| Parameter | Baseline Value | Elevated Value |
|---|---|---|
| Temperature (K) | 292 | - |
| Pressure (hPa) | 1000 | - |
| Solar Intensity (unitless) | 0 | 1 |
| OH (ppb) | 1e-6 | 1e-4 |
| NO (ppb) | 1.17e-6 | 0.53 |
| $NO_2$ (ppb) | 1e-4 | 0.01 |
| $NO_3$ (ppb) | 2.3e-4 | 0.02 |
| $HO_2$ (ppb) | 0.04 | 0.2 |
| $O_3$ (ppb) | 16.7 | 100 |
| $MO_2$ (ppb) | 0.1 | 0.2 |

of priority species yields under these conditions. There were 256 possible paths, represented by non-duplicate combinations of
the possible oxidants or nitrogen oxides. Temperature and pressure are other parameters that significantly influence isoprene
chemistry. However, these parameters are implicit to the graph as inputs to calculate rate constants. Thus, temperature and
pressure were not represented explicitly in the algorithm, leaving rate constants to be determined either through calibration or
through direct reaction analogues in the full mechanism. The default temperature and pressure for yield estimates were 292 K
and 1000 hPa respectively.

Using the yield estimation algorithm, a measure of the importance of each path was determined by evaluating the product
yields for a sequence of inputs designed to probe the sensitivity to each oxidant, nitrogen oxide, or photolysis. Each oxidant and
nitrogen oxide was assigned a baseline concentration or intensity, determined from atmospherically relevant ranges in which
rates of reactions involving each species were similar. For example, the baseline concentrations of $O_3$, OH, and $NO_3$ were set
such that the rate of reaction of isoprene with each oxidant or nitrogen oxide would be the same. Input sequences were created
in which the concentration of each oxidant and nitrogen oxide (or photolysis intensity) within the path is elevated, in turn,
roughly one order of magnitude above the baseline. Table 2 shows the input values used for the path importance algorithm. All
possible combinations of each high and low value were used as an input space, resulting in 256 different input conditions. The
goal in selecting input conditions was to find values that were relatively low and relatively high without biasing the algorithm
with extreme values. They do not represent the full range of values that each input takes. The AMORE-Isoprene mechanism
performs satisfactorily under more extreme conditions than those that were used as input conditions to the algorithm, but it
would be possible to create a mechanism optimized for a more extreme scenario using the AMORE algorithm. We conducted
a sensitivity test of the pathway importance algorithm to a select set of changes to the inputs shown in Table 2. Specifically,
elevated concentrations of OH, NO, and $NO_2$, were adjusted to reflect realistic upper values for these species. The results of
this test are given in Section S.13 and Table S.5.

For each hypothetical path, the yield of priority species from that path was determined by elevating the input values of the oxidants or nitrogen oxides in the path. A path was considered important if this process resulted in yield estimates that differed significantly from the baseline. Multi-step path yields were evaluated in comparison to paths with one less elevated oxidant or nitrogne oxide step. For example, the path {OH, HO$_2$, NO} was compared to the paths {OH, HO$_2$}, {OH, NO}, and {HO$_2$, NO}. If the yield of priority species differed significantly from all of the compared paths, then the path was deemed important.

The importance of each path was ranked in terms of the magnitude of difference in yield of the path to the least different shorter path. This method ensured that every component of the path was necessary to produce unique yields compared to the baseline. Equation 3 shows the importance metric used to choose the most important paths,

$$I = min\left(\sum_{i=1}^{N} \frac{abs(Y_{i,isop}^0 - Y_{i,isop}^x)}{max(Y_i, isop)}, \forall x \in \text{path list}\right) \tag{3}$$

where $I$ is the path importance, $N$ is the number of species in the important species list, $i$ is an individual important species in

the list, $Y_{i,isop}^0$ is the yield of species $i$ from isoprene for the path being measured, $Y_{i,isop}^x$ is the yield of species i from isoprene in path x, $max(Y_{i,isop})$ is the maximum yield obtained by species $i$ from the yields of all paths, and the path list is the set of paths with one less elevated oxidant or nitrogen oxide than the path being measured.

### 2.3.3   Pathway combination algorithm

Using this path analysis, the following 8 paths were identified and incorporated in the mechanism: {O$_3$}, {NO$_3$}, {NO$_3$,

HO$_2$}, {NO$_3$, HO$_2$, hv}, {NO$_3$, NO}, {OH}, {OH, HO$_2$}, {OH, NO}. The number of paths was chosen based on the desired mechanism size, but the paths were determined by the pathway importance algorithm above. In order to reduce the number of intermediates, paths were joined together such that any shared oxidant or nitrogen oxide within paths had a shared intermediate. For example, all paths involving OH were structured so that the first reaction was with isoprene and OH which then formed a shared intermediate. The reaction paths were algorithmically structured to share as many intermediates as possible. The

pathway combination algorithm started by grouping paths by a shared intermediate. For example, the paths {NO$_3$}, {NO$_3$, HO$_2$}, {NO$_3$, HO$_2$, hv}, and {NO$_3$, NO} all share a common NO$_3$ reaction step. There are instances in which there are multiple ways in which to group pathways. For example, {OH, NO} can either be grouped with other OH containing pathways or with NO pathways. There was no algorithmic way to prioritize these two options. This is an instance in which manual intervention is required to assign preference between pathway groupings. This can be done by simply choosing the order in

which the pathway reactions should occur. For example, choosing the order {OH, NO} would group this pathway with other OH pathways, whereas choosing the order {NO, OH} would group this pathway with other NO pathways.

     Once the groupings are formed, an initial reaction step is created in which isoprene reacts with the commonly shared oxidant or nitrogen oxide to form an intermediate shared by all of the pathways. For example, in the NO$_3$ pathway grouping, the reaction of isoprene with NO$_3$ is shared with all pathways, which subsequently branch from each other. Pathways that share

two oxidants or nitrogen oxides, such as {NO$_3$, HO$_2$} and {NO$_3$, HO$_2$, hv} share two intermediates. By grouping pathways by shared oxidants and nitrogen oxides, and creating sub-groupings for multiple shared oxidants and nitrogen oxides, the pathway

combination algorithm creates a reduced mechanism structure. This algorithm does not allow for the recombination of branched pathways, meaning that the resulting reduced graphs are necessarily trees. Figure S.1 demonstrates the combination of all of the identified paths to form the reduced mechanism structure.

### 2.3.4 Priority species yield determination

The yield of each priority species is measured for each path using the yield estimation algorithm (see Section 2.3.1). These yields are used as stoichiometric coefficients for the product terms of the terminal reaction of each path. All priority species are considered eligible as product terms of the terminal reaction of a given path. For each path, the terminal reaction is defined as the reaction in which no additional intermediates were produced. For example, the path {OH} contains the reaction of isoprene with OH to form isoprene hydroxy peroxy radical as an intermediate (reaction 1 in Table S.2). This pathway was then given a terminal reaction, involving the first order decomposition of the isoprene peroxy radical in order to produce the final priority oxidation products (reaction 2 in Table S.2). The stoichiometric coefficients of each oxidation product were the yields as estimated by the yield estimation algorithm.

This algorithm completed the automated portion of the mechanism development process. The fully automated mechanism is described in Table S.2. The assignment of reaction rate constants and species naming are discussed in the following section. The subsequent manual optimization process for direct implementation into 3-D atmospheric models is described in Section 2.4.

### 2.3.5 Rate parameter identification and species naming

Once the skeletal reduced mechanism was established, rate parameters and species names were identified manually. The first step was to identify any direct analogues between mechanisms in the reduced mechanism and known reactions (i.e., those in the Caltech full mechanism). There were many reactions with direct analogues, including all reactions involving isoprene. In these cases, the rate law and parameters assigned were identical to the original.

For reactions without direct analogues, the reaction was typified by the oxidant or nitrogne oxide involved. In the Caltech full mechanism, reaction rate laws with the same oxidant or nitrogen oxide tend to have a similar form and fall under a limited range of parameter values. Where there were multiple possible reaction forms, the most common form was chosen. After choosing the form of the rate laws, parameters were tuned by running box model simulations under conditions that favored the reaction being tested. The parameters were calibrated to match the concentration profiles of dominant products in comparison to the Caltech full mechanism. A list of all rate laws and parameters, their analogues, and the method of selection is given in Table S.3.

All species names listed in the AMORE-Isoprene mechanism were manually identified after the completion of the automated mechanism reduction process. As with the rate law selection process, the first step was to identify direct analogues in the full mechanism. Since the AMORE-Isoprene mechanism is highly reduced, all species with the exception of isoprene represent groups of species in the Caltech full mechanism. Thus, direct analogues were generally analogous groups of species. For species without a clear analogue, naming was based on the oxidants and nitrogen oxides that reacted to form the species. From

this information, a name was assigned based on the predicted functional groups present in the species. For CMAQ modeling, the naming convention is different for some species due to their prior existence in the model. Table S.4 gives each species name for this paper and for CMAQ, the analogues it represents, and the functional groups involved.

## 2.4 Manual mechanism optimization and evaluation

The algorithmically generated isoprene mechanism was manually optimized for use in the CMAQ modeling environment and evaluated for its performance compared to other reduced mechanisms. The optimization process was done using the F0AM box model (Section 2.4.1) and the CMAQ testing environment (Section 2.4.2), and the manual optimization process is described in Section 2.4.3. The graph theoretical framework helped inform our decisions in this process. For example, the conceptualization of the mechanism as a set of unique pathways connected by sequences of reactions, which is rooted in graph theory, helped us to categorize reactions and how adjustments to their parameters would impact end results under different testing conditions.

In the process of evaluating the mechanism, an error metric was developed and used for quantitative comparisons between mechanisms (Section 2.4.4). In the optimization and evaluation phase, the Caltech full mechanism was used as a baseline for comparison, along with experimental chamber data for further corroboration (Paulot et al. (2009)).

Higher priority was put on mechanism accuracy rather than retention of the original algorithmically generated mechanism structure. Thus, changes were made that deviated from the algorithmically generated mechanism, however the core components of the algorithmically generated mechanism, including a majority of the identified important paths, were retained, and the algorithmically generated mechanism provided an essential functional starting point from which to improve the final mechanism performance.

### 2.4.1 Box model testing

The Framework for 0-D Atmospheric Modeling (F0AM) (Wolfe et al. (2016)) was used to simulate isoprene mechanisms for the purpose of evaluating the AMORE-Isoprene mechanism. 0-D box model testing was done in two primary phases. The first phase was aimed at optimizing the AMORE-Isoprene mechanism. The Caltech full mechanism was taken as the most accurate mechanism for ground truth, and RACM2 was used as a benchmark for comparison. Simulated concentration profiles of key species such as $NO_x$, $HO_x$, IEPOX, $O_3$, and formaldehyde were analyzed in order to assess the AMORE-Isoprene mechanism. The goal was to match both the magnitude and form of each species concentration in the Caltech full mechanism. A detailed description of the matching process is provided in Section 2.4.3.

The second phase of box model testing involved quantitative comparisons between mechanisms for demonstration of the performance of the AMORE-Isoprene mechanism. The mechanism was tested in the F0AM environment alongside the Caltech full mechanism, RACM2 isoprene mechanism used in base CRACMM1, Carbon Bond 6 revision 3 (CB6r3), and the Caltech reduced isoprene mechanism. An error metric was created to determine the degree of matching between two concentration curves. This error metric was averaged over many species and conditions to create an overall mechanism error metric. Section 2.4.3 gives a detailed description of the error metric developed for this study.

**Table 3.** F0AM box model testing input conditions used for calculating the error metric and evaluating the AMORE-Isoprene mechanism. Bolded values represent species concentrations that were held constant. All other concentrations varied with time after initiation of the simulation.

| Species | Chamber Comparison | Low NO$_x$ | High NO$_x$ | High NO$_3$, Low hv | NO$_3$ | High O$_3$ |
|---|---|---|---|---|---|---|
| isoprene (ppb) | 92.5 | 10 | 10 | 10 | 10 | 10 |
| H$_2$O$_2$ (ppb) | 1660 | 200 | 200 | 0 | 100 | 200 |
| NO (ppb) | 1 | 0.5 | 5 | 2 | 1 | 0.5 |
| O$_3$ (ppb) | 0 | 0 | 0 | 0 | 0 | 100 |
| NO$_3$ (ppb) | 0 | 0 | 0 | **0.02** | **0.02** | 0 |
| hv (unitless) | 3.5 | 3.5 | 3.5 | 0.5 | 3.5 | 3.5 |

A set of six input conditions were devised to simulate the mechanisms. Given that isoprene oxidation is split into three main pathways of reaction with OH, NO$_3$, and O$_3$, these three pathways must be represented in the chosen testing conditions. Subsequent oxidation with NO is particularly important in the OH pathway, and low light conditions are important in the NO$_3$ pathway. Given this, the first five conditions were low NO$_x$, high NO$_x$, high O$_3$, high NO$_3$, and high NO$_3$ + low hv. The final input condition was set to simulate the chamber study of Paulot et al. (2009), allowing for the pairing of box model results to experimental results. In that study, H$_2$O$_2$ photolysis was used as the source of OH, and small amounts of NO$_x$ were measured as well. For all F0AM simulations, H$_2$O$_2$ was used as the source of OH (which allowed for OH to be a dynamic quantity), and NO was used as a source of NO$_x$. For ozone and NO$_3$ the concentrations were set directly. Due to the lack of NO$_3$ cycling and the resulting rapid decay of NO$_3$, NO$_3$ concentrations were held constant in for high NO$_3$ conditions in order to favor this pathway through the duration of the simulation. Temperature and pressure were held at 292 K and 1000 hPa for all conditions. This corresponds to low elevation warm conditions that are most relevant for isoprene chemistry. The rate of photolysis reactions are scaled by a unitless parameter labeled as hv. The value of this parameter was calibrated to match results of Paulot et al. (2009) chamber data for high photolysis conditions. Table 3 shows the inputs for each of the six conditions.

## 2.4.2 CMAQ modeling

CMAQ v5.3.3 (Appel et al. (2021)) with additional updates as in Place et al. (in prep) was used to conduct simulations over the northeastern U.S. for June through August 2018 (May 2-31 used as spinup) at 4km by 4km horizontal resolution. Baseline gas and aerosol-phase chemistry was specified by CRACMM version 1 (Pye et al., in prep) which uses the RACM2 representation of isoprene chemistry (Sarwar et al. (2013)). Additional simulations were conducted in which CRACMM's isoprene chemistry was replaced with AMORE-Isoprene. Meteorology was obtained from WRF v4.1.2 (Torres-Vazquez et al. (2022)) and processed through the Meteorology-Chemistry Interface Processor version 5 (Otte and Pleim (2010)). Boundary and initial conditions were mapped from previous work using CB6r3 (Torres-Vazquez et al. (2022)), and emissions were respeciated for CRACMM with additional updates for volatile chemical products (Seltzer et al. (2021)). Biogenic emissions were estimated with the Biogenic Emission Inventory System (BEIS) (Bash et al. (2016)) with M3dry (Pleim et al. (2019))

used for deposition. CMAQ output was compared to EPA Air Quality System (AQS) and other monitoring network data using
the Atmospheric Model Evaluation Tool (AMET) (Appel et al. (2011)). CRACMM was selected as a baseline mechanism due
to concurrent development of AMORE-Isoprene and the CRACMM mechanism for use in EPA research. CRACMM indicated
relatively consistent predictions of gas-phase ozone chemistry as other current mechanisms (Place et al. in prep), signifying
that the choice of CRACMM as the baseline mechanism for 3-D modeling was unlikely to confound the AMORE-Isoprene
results.

IEPOX has heterogeneous chemistry in CMAQ (reactive uptake leading to SOA) following Pye et al. (2013) with updates
in Pye et al. (2017) and Pye et al. (2022). The first generation isoprene organic nitrate heterogeneous heterogeneous chemistry
(leading to $HNO_3$ and gas-phase alcohols) was implemented in this work and is specific to AMORE (not in base CRACMM1).

In CMAQ, the species in AMORE undergo deposition. All species that were already present in the base CRACMM1 mech-
anism were treated the same as in CRACMM1. IPN and IPC were both wet deposited with Henry's law coefficients predicted
by OPERA (Mansouri et al. (2018)). In addition, the species were dry deposited using species-specific diffusivities, mesophyll
resistances, and LeBas molar volumes (Pye et al. (2017)).

### 2.4.3 Manual mechanism adjustment

In this section we discuss manual adjustments to the algorithmically generated mechanism. To make adjustments, we tested
the AMORE mechanism in box model simulations (Section 2.4.1) and 3D Chemical transport simulations (Section 2.4.2). The
testing process highlighted issues with the mechanism initially produced by the reduction algorithm that could be corrected
via manual adjustments. This process has informed future algorithm development, since the ultimate goal is to automatically
generate mechanisms which require no manual adjustment. The structural differences between the automated (labelled as
AMORE-NoAdjust) and manually adjusted (labelled as AMORE-final) mechanisms are shown in Figure 2. The corresponding
reaction numbers from Table 4 are shown in the AMORE-final structure. Reaction 13-17 are not shown in the structure because
they represent degradation schemes for end product species (IEPOX, IHN, and ISON), or are used for oxidant and nitrogen
oxide cycling and do not directly contribute to the production of priority species (reaction 13).

The first issue to be addressed was that, because of the DAG assumptions, mechanistic pathways were constrained to have
only forward reactions, and because of the pathway identification algorithm, each oxidant and nitrogen oxide was only able
to appear once. Adjustments were made to the mechanism to allow reversible reactions and repeat appearances of a single
oxidant or nitrogen oxide where there was a strong case for the adjustment based on the Caltech full mechanism. One of the
most important instances of this is the isoprene OH oxidation pathway. In this pathway, OH and $HO_2$ are the most important
oxidants as predicted by the AMORE algorithm, however, there is a reversible OH reaction (reaction 6 in Table 4) which plays
a significant role in the cycling of $HO_2$ and OH. This reversible reaction was added and was instrumental in improving the
accuracy of the AMORE-Isoprene mechanism. The change is shown in the {OH, $HO_2$} pathway of Figure 2. This reversible
reaction is absent from RACM2 but present in CB6r3. In order for the reversible reaction to terminate into final products, a
reaction of the second intermediate with OH was added. The addition of these two reactions did not change the overall nature

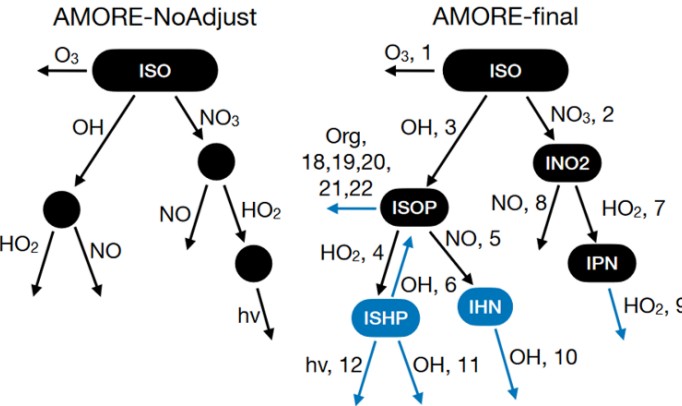

**Figure 2.** The original AMORE algorithmically generated mechanism prior to manual adjustment (left) and the final AMORE-Isoprene mechanism (right). Changes are highlighted in blue.

of the path {OH, HO$_2$} in terms of the oxidants present, but added necessary complexity to the dynamics of the path, resulting in more accurate product differentiation in OH dominant conditions.

390   It was also observed that NO$_x$ concentrations were relatively low compared to the Caltech full mechanism in low NO$_x$ regimes in which the OH, HO2 pathway was dominant. To ameliorate the lower NO$_x$ concentrations, an additional intermediate called IPC was created for the purpose of reacting with NO to create additional NO$_2$ and NO. This addition is shown in reaction 11 where IPC is a product, and reaction 13, where NO and NO$_2$ are cycled. The effect of this addition is to increase NO$_x$ under low NO$_x$ conditions, and thus increase ozone, leading to reduced ozone underestimation compared to the Caltech full

395   mechanism.

In addition to box model testing, 3D chemical transport modeling using CMAQ (Section 2.4.2) informed structural adjustments to the AMORE mechanism. These adjustments centered on the treatment of IHN (isoprene hydroxy nitrate, the intermediate of the reaction of ISOP (isoprene hydroperoxy radical) and NO (reaction 5 in Table 4, as part of the {OH, NO} pathway) in the mechanism. IHN was not initially identified as a priority species during algorithm development, and thus

400   was not included as an intermediate. Instead, the reaction of ISOP and NO contained no intermediates and led directly to the production of priority end products. However, it was determined that IHN should be given priority based on recent research highlighting its importance in NO$_x$ cycling (Vasquez et al. (2020)). Thus, IHN was added as an intermediate, and an additional decomposition reaction with OH was added (reaction 10 in Table 4). This decomposition reaction led to the production of IEPOX and isoprene nitrates, which were originally produced directly from the reaction of ISOP with NO. Thus, the {OH,

405   NO} pathway was expanded on by adding an additional OH reaction step for the decomposition of IHN. This change is shown in the {OH, NO} pathway with an addition of an OH reaction step in Figure 2. In addition to decomposition into other organics, IHN acts as a sink for NO$_x$. This was represented by the addition of reaction 16 in Table 4, which did not involve any oxidants or nitrogen oxides as reactants. It was observed that the reaction of IPC with NO (reaction 13 in Table 4) outcompeted IHN

**Table 4.** The AMORE-Isoprene mechanism reactions and rate constants. Mechanism specific species are listed in the text.

| # | Reaction | Rate Constant |
|---|---|---|
| 1 | ISO + O3 = 0.07 MACR + 0.189 MVK + 0.25 HO + 0.25 HO2 + 0.58 HCHO + 0.08 MO2 + 0.1 ACO3 + 0.09 H2O2 + 0.1 MACP + 0.461 MACR + 0.14 CO + 0.28 ORA1 + 0.15 OLT | $1.58E\text{-}14 \exp(-2000/T)$ cm$^3$ mol$^{-1}$ s$^{-1}$ |
| 2 | ISO + NO3 = INO2 + 0.3 HCHO + 0.3 NO2 + 0.3 ISON | $2.95E\text{-}12 \exp(-450/T)$ cm$^3$ mol$^{-1}$ s$^{-1}$ |
| 3 | ISO + HO = ISOP + 0.02 MO2 | $2.69E\text{-}11 \exp(390/T)$ cm$^3$ mol$^{-1}$ s$^{-1}$ |
| 4 | ISOP + HO2 = ISHP + 0.6 HO2 + 0.15 HCHO | $4.5E\text{-}13 \exp(1300/T)$ cm$^3$ mol$^{-1}$ s$^{-1}$ |
| 5 | ISOP + NO = 0.14 IHN + 0.7 HCHO + 0.44 MVK + 0.88 HO2 + 0.78 NO2 + 0.28 MACR + 0.021 GLY | $2.7E\text{-}12 \exp(350/T)$ cm$^3$ mol$^{-1}$ s$^{-1}$ |
| 6 | ISHP + HO = ISOP | $4.6E\text{-}12 \exp(200/T)$ cm$^3$ mol$^{-1}$ s$^{-1}$ |
| 7 | INO2 + HO2 = IPN + HO | $3.14E\text{-}14 \exp(580/T)$ cm$^3$ mol$^{-1}$ s$^{-1}$ |
| 8 | INO2 + NO = 0.2 ISON + 0.9 HCHO + 0.5 MGLY+ 0.8 MVK + 0.5 NO2 + HO2 + 0.1 MO2 | $9.42E\text{-}16 \exp(580/T)$ cm$^3$ mol$^{-1}$ s$^{-1}$ |
| 9 | IPN + HO2 = 0.2 ISON + 0.8 NO2 + 0.4 HCHO + 0.05 GLY + 0.1 MGLY + 0.4 MACR + HO2 + 0.94 MVK + 0.1 MO2 | $3.4E\text{-}11 \exp(390/T)$ cm$^3$ mol$^{-1}$ s$^{-1}$ |
| 10 | IHN + HO = ISON + HO + 0.2 IEPOX | $2.4E\text{-}7 \exp(580/T)$ cm$^3$ mol$^{-1}$ s$^{-1}$ |
| 11 | ISHP + HO = 0.05 IPC + 0.15 HCHO + 0.05 MGLY + 0.15 MACR + 0.02 GLY + 0.2 MVK + 0.4 NO2 + 0.58 IEPOX + 0.8 HO | $2.97E\text{-}11 \exp(390/T)$ cm$^3$ mol$^{-1}$ s$^{-1}$ |
| 12 | ISHP = 0.4 HCHO + 0.1 MGLY + 0.06 ACO3 | Photol(HCHO_RAD_RACM2) s$^{-1}$ |
| 13 | IPC + NO = 0.35 NO2 + 0.8 NO | $1e\text{-}10$ cm$^3$ mol$^{-1}$ s$^{-1}$ |
| 14 | ISON + HO = CO + 0.12 NO2 | $5e\text{-}11$ cm$^3$ mol$^{-1}$ s$^{-1}$ |
| 15 | ISON + NO3 = CO | $2e\text{-}14$ cm$^3$ mol$^{-1}$ s$^{-1}$ |
| 16 | IHN = HNO3 | $2.3e\text{-}5$ s$^{-1}$ |
| 17 | IEPOX + HO = HO | $5E\text{-}11 \exp(-400/T)$ cm$^3$ mol$^{-1}$ s$^{-1}$ |
| 18 | ISOP + MO2 = HO2 + 1.31 HCHO + 0.159 MACR + 0.250 MVK + 0.250 MOH + 0.250 ROH + 0.023 ALD + 0.018 GLY + 0.016 HKET | $3.4E\text{-}14\exp(221/T)$ cm$^3$ mol$^{-1}$ s$^{-1}$ |
| 19 | ISOP + ACO3 = 0.5 HO2 + 0.5 MO2 + 1.048 HCHO + 0.219 MACR + 0.305 MVK + 0.5 ORA2 | $8.4E\text{-}14 \exp(221/T)$ cm$^3$ mol$^{-1}$ s$^{-1}$ |
| 20 | ISOP + APIP2 = 0.96 HOM + 0.48 ROH + 0.48 HCHO + 0.48 MVK + 0.48 HO + 0.48 HO2 + 0.04 ELHOM | $1e\text{-}10$ cm$^3$ mol$^{-1}$ s$^{-1}$ |
| 21 | ISOP + APINP2 = 0.96 HOM + 0.48 ROH + 0.48 HCHO + 0.48 MVK + 0.48 NO2 + 0.48 HO2 + 0.04 ELHOM | $1e\text{-}10$ cm$^3$ mol$^{-1}$ s$^{-1}$ |
| 22 | ISOP + LIMNP2 = 0.96 HOM + 0.48 ROH + 0.48 HCHO + 0.48 MVK + 0.48 NO2 + 0.48 HO2 + 0.04 ELHOM | $1e\text{-}10$ cm$^3$ mol$^{-1}$ s$^{-1}$ |

for NO, and thus the yield of IPC (reaction 11) was changed from 0.3 to 0.05 from ISHP. This change came at the expense of NO$_x$ cycling under low NO$_x$ conditions, however, it was observed that simulated NO$_x$ levels were largely the same between AMORE-Isoprene and the base CRACMM mechanism, suggesting that this adjustment would be a net benefit to the overall performance.

Additional reactions of the OH pathway with organic radicals (methyl radical, peroxyacetyl radical, and lumped terpene radicals) were added directly from the RACM2 mechanism. They were not identified as important by the AMORE mechanism, likely a result of the inputs chosen, but came at little additional computational cost since they did not require the addition of any intermediates. These added organic radical reactions allowed for product differentiation in environments where organic radical concentrations are significant.

Finally, the {NO$_3$, HO$_2$, hv} path was determined to be unnecessary due to the relatively small amount of flux carbon directed to it. Instead, the paths {NO$_3$, HO$_2$, HO$_2$} and {OH, HO$_2$, hv} were used in its place. The {NO$_3$, HO$_2$, HO$_2$} path was a variant on the existing {NO$_3$, HO$_2$} path. The {OH, HO$_2$, hv} path was added to represent any potential variation attributed to photolysis in the low NO$_x$ regime. All of the above changes are shown in Figure 2.

The stoichiometric coefficients of the products in the reduced mechanism were initially assigned based on the estimates given by the yield estimation algorithm, and then optimized manually. Notably, stoichiometric coefficients for oxidant and nitrogen oxide species as products, although clearly important, were not treated in the algorithm due to their implicit representation in the mechanism graph. Thus, while oxidants and nitrogen oxides were included in the algorithm as reactants, they were omitted as products, which reduced their overall accuracy. In particular, the relationship between HO$_x$, NO$_x$ and O$_3$ is very sensitive to changes in the isoprene mechanism and is important for determining yields of many other species. These oxidants and nitrogen oxides are of a high order of importance for mechanism accuracy, and so the manual adjustment of their presence in the mechanism was critical. Two clear examples of this are shown in Figure 3. Prior to the adjustment shown in Figure 3.a, HO$_2$ was significantly reduced in the AMORE-Isoprene mechanism under low NO$_x$ conditions. It was observed that reaction 4 (Table 4) involving isoprene hydroxy peroxy radical (ISOP) was the main sink for HO$_2$, and by adding HO$_2$ to the product term, the accuracy was significantly increased. This original discrepancy likely reflects a cycling achieved by multiple reactions in the full mechanism that it was not possible to include in the reduced mechanism. Adding HO$_2$ to the product term was the only available way to have good agreement with the full mechanism. The reaction of isoprene hydroxy peroxy radical (ISOP) with NO was another reaction for which oxidant and nitrogen oxide cycling was very impactful. The addition of NO$_2$ and HO$_2$ to the products of reaction 5 (Table 4) was used to improve the accuracy of the AMORE-Isoprene mechanism. A demonstration of the adjustment improvements is shown in Figure 3.b. Further tests of the adjustments in Figure 3 are given in section S.17.

All of the above adjustments were motivated by a clear improvement in mechanism performance that accompanied the change. See sections S.8 and S.9 and Tables S.3 and S.4 for description of rate constants and species names.

All product stoichiometric coefficients were optimized for the accuracy of all priority species in a method similar to that shown for oxidants and nitrogen oxides in Figure 3.

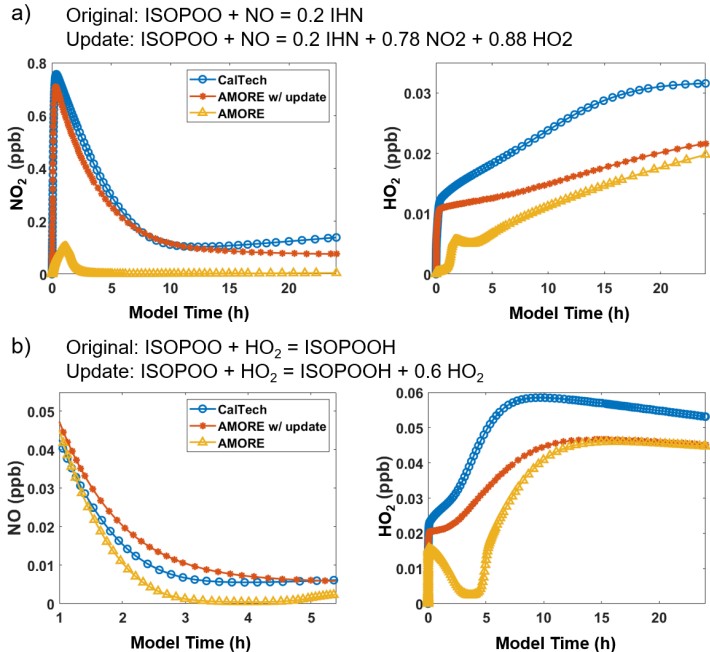

**Figure 3.** Box model simulations (T = 292 K, p = 1000 hPa) showing the improvement in performance of the AMORE mechanism for $HO_2$ and $NO_x$ after adding these species to the products of reactions 4 (b) and 5 (a). The original and updated reactions are shown above the plots. Inputs are a) 200 ppb $H_2O_2$, 1 pbb NO, 10 ppb isoprene, and moderate photolysis conditions (F0AM photolysis parameter = 1). b) 200 ppb $H_2O_2$, 1 pbb NO, 10 ppb isoprene, and high photolysis conditions (F0AM photolysis parameter = 3.5).

### 2.4.4 Mechanism error metric

For inter-comparison of reduced mechanisms, full mechanisms, and experimental data, it was necessary to devise an accuracy metric based on the priority species and other measurable parameters. In the case of the isoprene mechanism, we focus on
atmospheric oxidants and nitrogen oxides, organic aerosol, and other pollutants, namely formaldehyde and ozone.

In order to measure these parameters and create an accuracy metric, three steps were taken. The first was to define an error function for comparing the concentration of a species between two mechanisms in a box model simulation. The second was to determine the set of input conditions needed to capture the desired range of performance in the mechanism. The third step was to average errors across species and conditions in order to come up with a final metric. The error function for the comparison of
concentration profiles of one species between two mechanisms formed the basis of the accuracy metric. The goal was to devise an error function which is bounded, so the natural choice was to normalize the error. In addition, for the purpose of averaging, the error function needed to always be positive to avoid canceling out errors. From this an error metric was defined shown in Equation 4.

$$E = \frac{\int_{t_0}^{t_f} abs(T(t) - R(t))dt}{\int_{t_0}^{t_f} max(T(t), R(t))dt} \tag{4}$$

Where $E$ is the error, $t_0$ is the initial time, $t_f$ is the final time, $T(t)$ is the concentration profile being tested and $R(t)$ is the reference concentration profile. This concentration error metric ranges from 0 to 1 where 0 is no error and 1 is infinite error. Figure S.3 illustrates the behavior of the error metric for a sample set of profiles.

Although many important species are tracked in the isoprene mechanism, not all species contribute equally to observable parameters. A weighting scheme was devised to capture the relative importance of some species over others. The three main
groupings that were included in the weighting scheme were oxidants and nitrogen oxides, priority pollutants, and isoprene SOA species. Each grouping was given an equal contribution to the overall error. The primary oxidant and nitrogen oxide species are OH, $HO_2$, NO and $NO_2$. $NO_3$ is not involved in any significant cycles and is excluded from the oxidant and nitrogen oxide weighting scheme, but still participates in the mechanism. The organic oxidants methyl radical and peroxyacetyl radical are of lesser importance than the primary oxidants and nitrogen oxides and are given a lower weighting. NO, $NO_2$, OH and $HO_2$
are all given a 7% weighting for the overall accuracy. The methyl radical and peroxyacetyl radical are given a weighting of 2.5% each for a total of 33% for oxidants and nitrogen oxides. Ozone and formaldehyde are classified as pollutants, and both are given a weighting of 17% for a total weighting of 34%. The formaldehyde error is multiplied by the fraction of maximum formaldehyde concentration for a given input condition over the average maximum formaldehyde concentration over all input conditions. This gives formaldehyde more weighting as its relative concentration increases.

According to Bates and Jacob (2019), average isoprene SOA contribution is divided up into 33% IEPOX, 30% isoprene nitrates, 30% tetrafunctional isoprene compounds, 2.5% glyoxal, and 4.5% other. Most small isoprene mechanisms exclude tetrafunctional compounds, leaving IEPOX as 50% contribution, 45% isoprene nitrates, and 4.5% glyoxal. As with formaldehyde, each of these are scaled relative to their average maximum concentration. Thus, in our calculations SOA contributes 33% to the total accuracy, with IEPOX contributing 16.5%, isoprene nitrates contributing 15% and glyoxal contributing 1.5%.
Isoprene is omitted from the error metric, since its error is represented by the accuracy in the other parameters. Methyl vinyl ketone, methacrolein, peroxyacetyl nitrate, and methyl glyoxal were omitted from accuracy metric due to their relatively lower importance compared to the other species, and their coupling to species already present in the error metric. However, these four species are represented in AMORE-Isoprene, and tables on the performance of each mechanism with respect to these species and all other important species can be found in section S.10. Table 5 shows each species and its contribution to the total error
metric.

The error metric is calculated by running box model simulations of the Caltech mechanism and the test mechanism in all six conditions, then calculating each individual species error and averaging them using the weights shown in Table 5 and then averaging between each of the six conditions to arrive at a single value. The error metric ranges from 0 to 1, with lower values corresponding to less error. This allows for the numerical comparison of various isoprene mechanisms to the Caltech
full mechanism.

**Table 5.** Species used in the calculation of the mechanism error metric and their corresponding weight.

| Species | Fractional Contribution |
|---|---|
| OH | 0.07 |
| $HO_2$ | 0.07 |
| NO | 0.07 |
| $NO_2$ | 0.07 |
| Methyl radical ($MO_2$) | 0.025 |
| Peroxyacetyl radical ($ACO_3$) | 0.025 |
| HCHO | 0.17 |
| $O_3$ | 0.17 |
| IEPOX | 0.165 |
| Isoprene Nitrates | 0.15 |
| Glyoxal | 0.015 |

## 3    Results and discussion

The final AMORE-Isoprene mechanism consists of 9 species and 22 reactions. A full table of the reactions is shown in Table 4. The 9 isoprene species were: isoprene (ISO), isoprene hydroxy peroxy radical (ISOP), isoprene hydroxy peroxide (ISHP), isoprene nitrooxy peroxy radicals ($INO_2$), isoprene hydroxy nitrates (IHN), the lumped species IPC and IPN, isoprene epoxydiol (IEPOX), and lumped multifunctional isoprene nitrates (ISON). IPC and IPN are named based on the reactions they participate in, but they have no true analogues in the full mechanism, as they are used primarily to expand the range of outputs and cycle oxidants and nitrogen oxides. In the following sections, AMORE-Isoprene's performance will be compared in Box model simulations to the Caltech full mechanism (Section 3.1), compared to Chamber data (Section 3.2), and compared to the CRACMM-baseline mechanism in CMAQ simulations (Section 3.3).

## 3.1    Ambient box model simulations

Using F0AM box model simulations and the error metric defined in Section 2.4.4, we were able to demonstrate the high accuracy of the AMORE-Isoprene mechanism relative to other mechanisms of similar size. Formaldehyde and $HO_2$ were chosen as exemplary species for visual comparison, as they demonstrate the high performance of AMORE-Isoprene relative to other isoprene mechanisms.

Figure 4 shows the concentration of $HO_2$ under the six conditions listed in Table 3. For $HO_2$, AMORE-Isoprene has stronger agreement with the Caltech full mechanism than the RACM2 isoprene mechanism. In low $NO_x$ conditions, the steady state concentration of $HO_2$ was 0.054 ppb for the Caltech full mechanismm, 0.045 ppb for the AMORE-Isoprene mechanism, 0.042 ppb for the CB6r3 mechanism, and 0.026 ppb for the RACM2 mechanism. In high $NO_x$ conditions, all mechanisms had similar concentrations of $HO_2$. In high $O_3$ conditions, steady state concentration of $HO_2$ was 0.05 ppb for the Caltech full

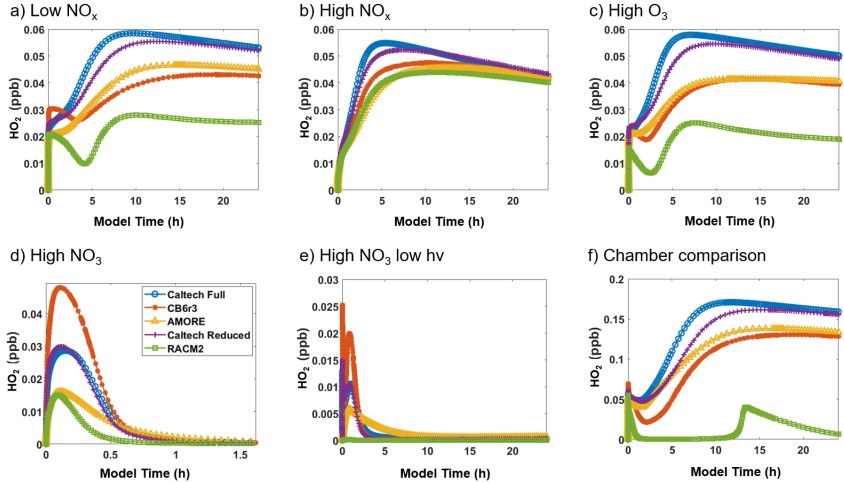

**Figure 4.** Box model predictions of $HO_2$ from multiple mechanisms (292 K and 1000 hPa) for the following conditions: a) Low $NO_x$ b) High $NO_x$ c) High $O_3$ d) High $NO_3$ e) High $NO_3$, low hv, and f) Paulot et al. (2009) Chamber.

mechanism, 0.04 ppb for AMORE-Isoprene, 0.04 ppb for CB6r3, and 0.02 ppb for RACM2. Under high $NO_3$ concentrations, steady state $HO_2$ concentrations were low for all mechanisms, and the peak concentration was 0.029 ppb for the Caltech full mechanism, 0.017 ppb for AMORE-Isoprene, 0.049 ppb for CB6r3, and 0.016 ppb for RACM2. In all cases, the steady state and/or peak concentrations were closer to the Caltech full mechanism for AMORE-Isoprene than for RACM2. Simulated $HO_2$ concentration profiles are similar between AMORE-Isoprene and CB6r3, though under high $NO_3$ conditions, AMORE-Isoprene tends to underestimate $HO_2$ whereas CB6r3 tends to overestimate $HO_2$. As expected, the Caltech Reduced plus mechanism has strong agreement with the Caltech Full mechanism for $HO_2$.

Figure 5 shows the simulated concentration of formaldehyde under the six conditions listed in Table 3. For formaldehyde, AMORE-Isoprene consistently outperforms RACM2. For example, in low $NO_x$ conditions, the peak formaldehyde concentration was just over 3 ppb for the Caltech full mechanism, 2.6 ppb for the AMORE-Isoprene mechanism, 2.1 pbb for the CB6r3 mechanism, and 0.6 ppb for the RACM2 mechanism. In addition, the mechanism performs similarly to CB6r3. The accuracy tends to be higher than CB6r3 for the first few hours of the simulation, with less accuracy later in the simulation due to a steeper decay of formaldehyde concentration. This pattern can be seen clearly in Figure 5.a. These differences arise from the sources of formaldehyde, where AMORE-Isoprene produces more formaldehyde from first generation oxidation intermediates, whereas CB6r3 has more contribution from later generation oxidation intermediates. Formaldehyde is overestimated in high $NO_3$ conditions, though other small mechanisms significantly underestimate formaldehyde under the same conditions. This difference is primarily due to the slower rate of decay of formaldehyde in high $NO_3$ conditions coupled with high initial production from the oxidation of isoprene. In contrast, the Caltech full mechanism displays a slower rate of formaldehyde production, as it is spread out over many more oxidation reactions at longer time scales. The other small reduced mechanisms

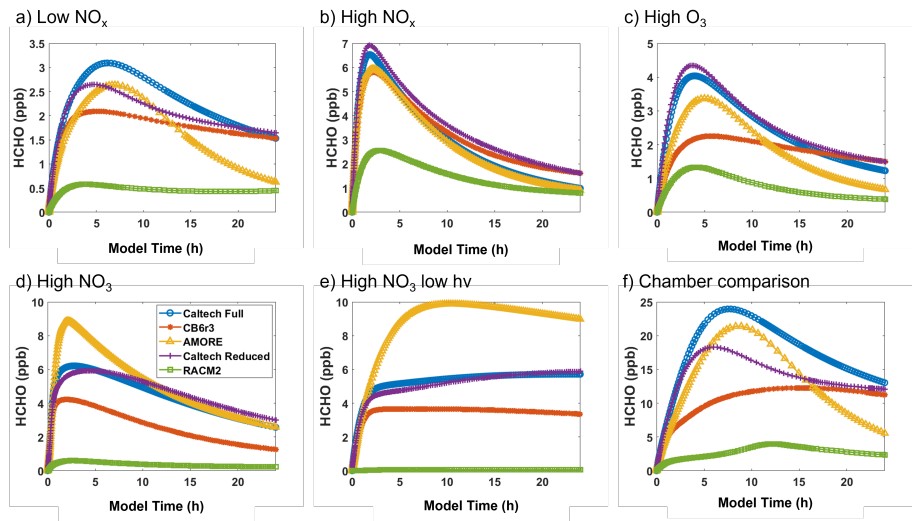

**Figure 5.** Box model predictions of formaldehyde from multiple mechanisms (292 K and 1000 hPa) for the following conditions: a) Low $NO_x$ b) High $NO_x$ c) High $O_3$ d) High $NO_3$ e) High $NO_3$, low hv, and f) Paulot et al. (2009) Chamber.

(CB6r3 and RACM2) have lower formaldehyde production, due to fewer formaldheyde production pathways under high $NO_3$

conditions, with RACM2 having over an order of magnitude lower formaldehyde concentrations.

    Figure 6 shows the simulated concentration of the hydroxyl radical under the six conditions listed in Table 3. The AMORE-Isoprene mechanism performs similarly to other highly reduced mechanisms. As with other small mechanisms, AMORE-Isoprene is biased low compared to the full Caltech mechanism. Under low $NO_x$ conditions, the AMORE-Isoprene mechanism has near equal behavior to CB6r3 and RACM2 at short time frames and has a more accurate steady state value at longer time

frames. At high $NO_x$, the RACM2 mechanism is the most accurate small mechanism, with AMORE-Isoprene having close but slightly lower OH concentrations. At high $O_3$, AMORE-Isoprene has the closest agreement with the Caltech full mechanism. The Caltech Reduced plus mechanism has strong agreement with the full mechanism at all tested conditions, as would be expected. The main reason for the discrepancy in between AMORE-Isoprene and the Caltech full mechanism in hydroxyl radical concentrations is that the Caltech full mechanism has a greater quantity of intermediate species which produce and

consume the hydroxyl radical. On balance, this leads to slightly higher hydroxyl radical concentrations, and given that the AMORE-Isoprene mechanism is a much smaller mechanism, there are limitations to the extent that this can be corrected. This is further evidenced by the fact that the other small mechanisms have equally low biased hydroxyl radical concentrations. Overall, the AMORE-Isoprene mechanism performs consistently well at predicting OH concentrations, and is in line with similarly sized mechanisms in this regard.

In addition to these plots, a quantitative comparison was made between each of the mechanisms tested based on the overall mechanism error metric defined in Section 2.4.4. Figure 7 shows the mean accuracy of AMORE-Isoprene for a selection of species in each of the six simulation conditions. Lower values correspond to higher accuracy. The AMORE-Isoprene mecha-

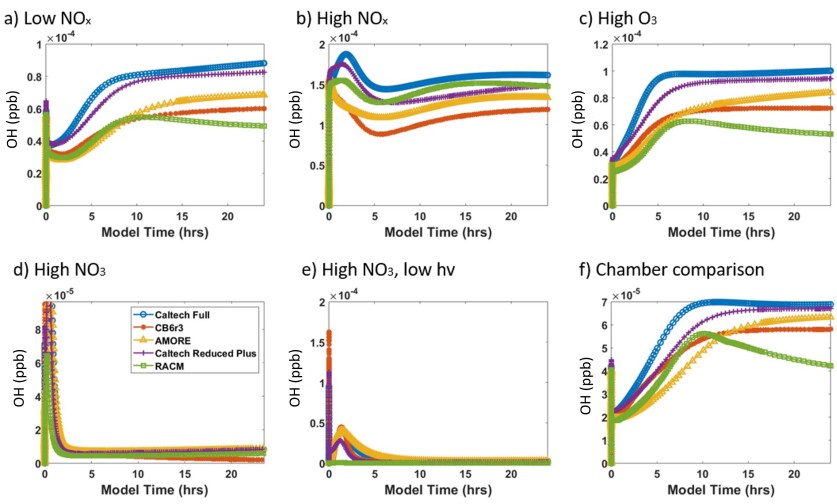

**Figure 6.** Box model predictions of the hydroxyl radical from multiple mechanisms (292 K and 1000 hPa) for the following conditions: a) Low NO$_x$ b) High NO$_x$ c) High O$_3$ d) High NO$_3$ e) High NO$_3$, low hv, and f) Paulot et al. (2009) Chamber.

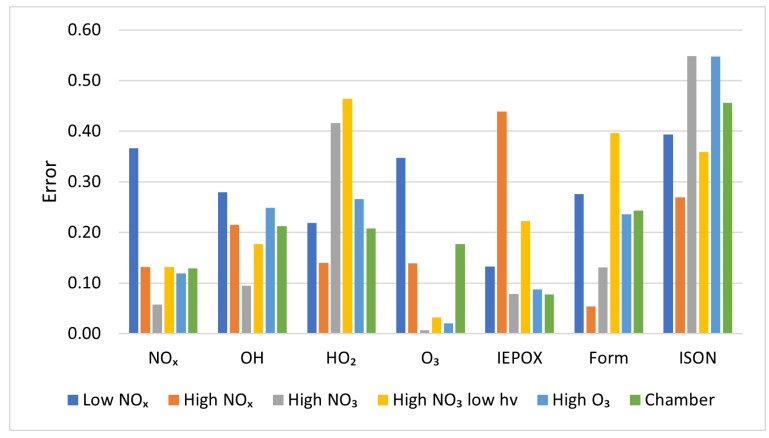

**Figure 7.** Measured error (Equation 4) of AMORE-Isoprene mechanism under six different conditions for seven select species groups. Errors are averaged between species for multiple-species groups.

nism shows very high accuracy under all conditions, and performed the best under high NO$_x$, high O$_3$, and in comparison to the Chamber data from Paulot et al, which is relatively low NO$_x$, and OH oxidation dominates.

Table 6 shows the overall error of each reduced mechanism as defined in Section 2.4.4 with species weightings described in 5. The numerical errors shown in the table represent weighted averages of the error across multiple priority species and the six conditions shown in Table 3. As a result, this error metric quantifies the overall performance of the mechanism in box model simulations, The results show that AMORE-Isoprene performs very well compared to mechanisms of a similar size. AMORE-

**Table 6.** Total Error (individual species error described in Equation 4, species weighting shown in Table 5, further discussion in Section 2.4.4) and mechanism size for four reduced isoprene mechanisms with the Caltech full mechanism as a basis of comparison. Individual species error shown averaged over the six tested conditions.

|  | AMORE | Caltech Reduced Plus | RACM2 | CB6r3 |
|---|---|---|---|---|
| Species | 12 | 131 | 9 | 10 |
| Reactions | 22 | 220 | 12 | 17 |
| Total Error | 0.17 | 0.13 | 0.44 | 0.3 |
| O3 | 0.12 | 0.02 | 0.15 | 0.12 |
| NO | 0.12 | 0.06 | 0.22 | 0.28 |
| NO2 | 0.19 | 0.08 | 0.38 | 0.42 |
| HO | 0.20 | 0.17 | 0.44 | 0.30 |
| HO2 | 0.29 | 0.11 | 0.67 | 0.29 |
| NO3 | 0.36 | 0.09 | 0.47 | 0.25 |
| ISOP | 0.14 | 0.06 | 0.18 | 0.11 |
| IEPOX | 0.17 | 0.12 | 0.60 | 0.27 |
| HCHO | 0.22 | 0.11 | 0.79 | 0.30 |
| MO2 | 0.53 | 0.20 | 0.59 | 0.56 |
| ACO3 | 0.56 | 0.27 | 0.72 | 0.44 |
| PAN | 0.53 | 0.21 | 0.85 | 0.52 |
| ISOPN | 0.43 | 0.26 | 0.61 | 0.77 |
| GLY | 0.64 | 0.60 | 0.86 | 0.57 |
| MGLY | 0.63 | 0.14 | 0.79 | 0.23 |

Isoprene has an error of 0.17, which is much lower than that of CB6r3 (0.3) and RACM2 (0.44), and very close to Caltech Reduced Plus (0.13) which is a significantly larger mechanism. The main drivers of the low error for the AMORE-Isoprene mechanism are oxidant, nitrogen oxide, IEPOX, and formaldehyde concentrations. For example, the average error for IEPOX is 0.17 for AMORE-Isoprene compared to 0.27 for CB6r3 and 0.60 for RACM2. The average error for $NO_x$ species is 0.16 for AMORE-Isoprene, 0.29 for RACM2 and 0.35 for CB6r3. For $HO_x$ species, the average error is 0.25 for AMORE-Isoprene, 0.55 for RACM2, and 0.29 for CB6r3. For formaldehyde, the average error is 0.22 for AMORE-Isoprene, 0.79 for RACM2, and 0.3 for CB6r3. The error tables for each mechanism can be found in section S.14 and Table S.6. Additional box model plots can be found in section S.15 and S.18, and Figures S.15-20. These results validate the AMORE reduction process as a useful method of mechanism reduction, and demonstrate that small mechanisms can retain significant accuracy compared to a much larger reference mechanism.

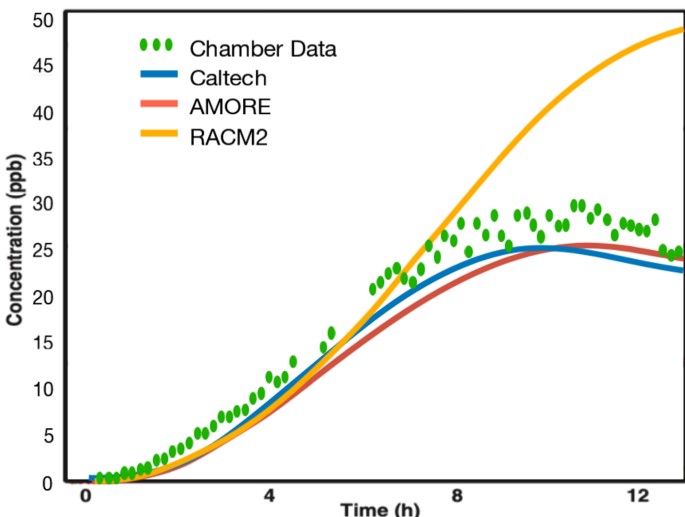

**Figure 8.** IEPOX concentration comparison between chamber data from Paulot et al. (2009) and F0AM box model simulations using the reported chamber conditions. The Caltech full mechanism closely matches the measured values, as does the AMORE-Isoprene mechanism.

## 3.2 Chamber box model simulations

In order to determine the accuracy of the Caltech full mechanism, which was augmented in this work, Chamber data was used for comparison. The data comes from Paulot et al. (2009), and contains concentration profiles for isoprene, isoprene hydroxy peroxides (ISHP), and IEPOX. The conditions of the chamber study were replicated using the F0AM box model to determine the accuracy of the Caltech full mechanism and the reduced isoprene mechanisms. As expected, the Caltech Full mechanism matched the concentrations of all measured species from the chamber study.

Figure 8 shows the results for IEPOX. The AMORE-Isoprene mechanism is in good qualitative and quantitative agreement with the Caltech full mechanism and the Paulot et al. (2009) chamber data concentration profile for IEPOX. The peak IEPOX concentration is roughly 27 ppb according to the chamber data, compared to 25 ppb for the Caltech full mechanism, 25 ppb for the AMORE-Isoprene mechanism, and 50 ppb for the RACM2 mechanism. In addition, the timing of the peak matches closely with the chamber data. The chamber data IEPOX peak occurs at around 10.5 hours, compared to 9.5 hours for the 570 Caltech full mechanism, 10.5 hours for the AMORE-Isoprene mechanism, and over 12 hours for the RACM2 mechanism. Model evaluation with chamber data is particularly important for this species because (a) it is a key species for SOA formation from isoprene and (b) relatively few ambient meaurements of IEPOX exist for validation.

## 3.3 CMAQ Testing

When AMORE-Isoprene is included in the CRACMM scheme of CMAQ (CRACMM1AMORE), improvements in $O_3$ and 575 formaldehyde bias are observed compared to AQS ambient observations. Figure 9 shows the bias of CRACMM-baselineline

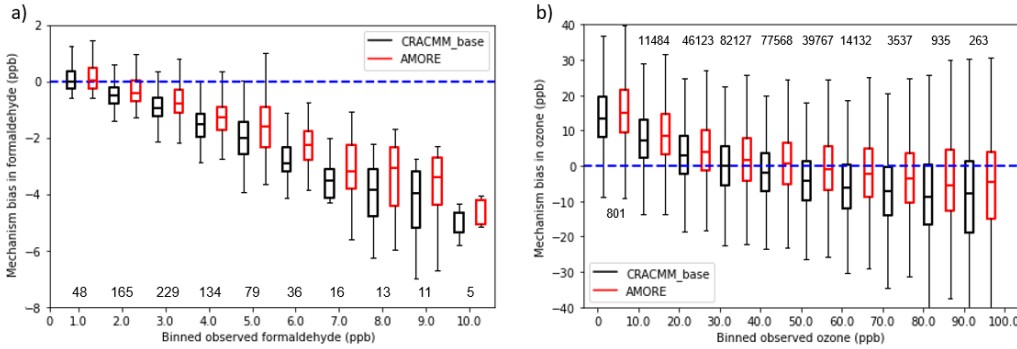

**Figure 9.** Binned mean bias of AMORE-Isoprene and baseline CRACMM for (a) formaldehyde and (b) ozone compared to AQS data for the Northeast U.S. during summer 2018. Numbers for boxplots indicate the number of data points in each observed range.

and AMORE-Isoprene compared to AQS data. Both formaldehyde and ozone observations were underestimated by CMAQ, particularly at higher concentrations of each species. AMORE-Isoprene predicted higher ozone and formaldehyde than CRACMM-baseline, thereby reducing the bias for both species.

The mean bias of formaldehyde decreased by 0.27 ppb with the implementation of AMORE-Isoprene, for all formaldehyde
concentrations. The CRACMM-baseline simulation tended to underestimate formaldehyde, especially at increasing formalde-
hyde concentrations. In all concentrations, AMORE-Isoprene increased the simulated formaldehyde concentration by roughly
0.25 pbb. This is in line with the box model simulations shown in Figure 5, where AMORE-Isoprene (yellow) had consistently
higher formaldehyde concentrations than the RACM2 mechanism (green), which is used in the CRACMM-baseline isoprene
mechanism, under a wide range of conditions.

The mean bias of ozone for concentrations above 50 ppb during the daytime was decreased by 3.4 ppb with the imple-
mentation of AMORE-Isoprene. AMORE-Isoprene slightly increased the bias at low ozone concentrations, because ozone
is overpredicted at low concentrations. At higher concentrations, where health implications are presumably more serious,
AMORE-Isoprene yielded significantly higher accuracy. AMORE-Isoprene generally tended to increase ozone concentrations
by roughly 2-3 ppb for all ozone concentrations. AMORE-Isoprene tends to have higher ozone concentrations and better
agreement with the Caltech full mechanism than RACM2 in low $NO_x$ box model simulations. Thus, the difference may be
attributable a higher prevalence of low $NO_x$ conditions.

The CMAQ implementation also included heterogeneous chemistry for IEPOX and first generation isoprene organic nitrates,
and deposition for all species. These processes, while not included in our box models, did not significantly impact the overall
performance of the mechanism, as OC values were similar between AMORE-Isoprene and the base CRACMM1 mechanism
(see section S.16). No significant changes were identified for other observed species such as $NO_y$, and $HNO_3$ (section S.16).
The CMAQ model runtime did not increase substantially with AMORE-Isoprene.

## 4 Conclusions

We have developed a new reduced isoprene oxidation mechanism for application in large-scale atmospheric models, using a novel, semi-automated, graph-theory based approach. Rigorous testing has demonstrated that the AMORE-Isoprene mechanism's performance is very good for its size, with improved accuracy compared to CB6r3 and RACM2.

A small, accurate isoprene oxidation mechanism would improve the performance of many large-scale models, as we have demonstrated with CMAQ-CRACMM1AMORE simulations, where there was a noticeable improvement in both ozone and formaldehyde bias. In the future we plan additional testing of AMORE-Isoprene in other chemical transport models to characterize the impacts of this mechanism more broadly.

During the algorithmic and manual adjustment process, several useful concepts were developed. First, for a small number of desired measurable outputs, small mechanisms can reach high levels of accuracy if properly structured and optimized. Second, optimization of oxidants and nitrogen oxides, which are highly coupled to the isoprene mechanism, takes precedence over optimization of other species, since inaccuracies in coupled species ultimately propagate to uncoupled species. In addition, the observation that methods reliant on removing aspects of the full mechanism would not work for this application was very important. The path-based approach we have developed to "summarize" the mechanism may be a more sensible starting point for reduction of other atmospheric reaction networks as well.

The AMORE-Isoprene mechanism demonstrates that there is significant potential advantage in the use of algorithms for model reduction. Additional development, informed by the experiences of this study, is underway to more fully automate the model reduction process and further reduce the need for manual adjustments. Future work will extend this work to application to reduction of a wide range of atmospheric chemical mechanisms in addition to the isoprene oxidation mechanism.

*Code availability.* CMAQv5.3.3 is available at https://github.com/USEPA/CMAQ and archived at doi: 10.5281/zenodo.5213949. The exact CMAQ code used in this work and CMAQ output is available at doi: 10.23719/1527975.

Code and data for the AMORE algorithm is available at https://github.com/fcw2110/AMORE_supplementary_files and archived at doi: 10.5281/zenodo.7106505.

*Author contributions.* All authors contributed to writing the manuscript. Wiser, Sen and McNeill developed the model reduction algorithm. Wiser, Place, Sen, Pye, and McNeill developed the reduced model. Wiser, Place, and Pye performed simulations. CRACMM was conceived by Pye. Research was conceived by Wiser, McNeill, Sen, Westervelt, Henze, and Fiore.

*Competing interests.* The authors declare no competing interests are present.

*Acknowledgements.* This publication was developed under Assistance Agreement No. 84001301 awarded by the U.S. Environmental Protection Agency to McNeill, Westervelt, Henze, and Fiore. The views expressed in this article are those of the authors and do not necessarily represent the views or policies of the U.S. Environmental Protection Agency.

EPA does not endorse any products or commercial services mentioned in this publication.

We would like to thank Dr. Kelvin Bates and Dr. Glenn Wolfe for helpful discussions. We thank Jon Pleim, Ana Torres-Vazquez, and Christine Allen for assistance with CMAQ simulation inputs.

This work was supported in part by the U.S. Environmental Protection Agency Office of Research and Development. This research was also supported in part by an appointment to the U.S. Environmental Protection Agency (EPA) Research Participation Program administered by the Oak Ridge Institute for Science and Education (ORISE) through an interagency agreement between the U.S. Department of Energy (DOE) and the U.S. Environmental Protection Agency. ORISE is managed by ORAU under DOE contract number DE-SC0014664. All opinions expressed in this paper are the author's and do not necessarily reflect the policies and views of US EPA, DOE, or ORAU/ORISE.

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
