# Peer review of "AMORE-Isoprene v1.0: A new reduced mechanism for gas-phase isoprene oxidation"

_Geoscientific Model Development, 2022_

## Author Comment (AC1)

Paper and Comments: GMDD - AMORE-Isoprene v1.0: A new reduced mechanism for gas-phase isoprene oxidation (copernicus.org)

We are grateful to the referees for their helpful comments, and for the opportunity to improve the manuscript based on their input. We have copied the reviewer comments below and we respond inline (in blue with changes to the main documents *italicized*)

**Reviewer 1**

This study presents the development of a new reduced isoprene oxidation scheme for application in a large-scale atmospheric model, using a benchmark state-of-the-science full description of the isoprene chemistry as a starting point, utilising a novel graph-theory based approach. The mechanism is then optimised and evaluated against its benchmark and other reduced schemes specifically designed for use in US regulatory models as well as limited chamber data, in box models as well as being incorporated into the US EPA Community Multiscale Air Quality modelling system (CMAQ v5.3.3) and evaluated against NE US air quality data.

This interesting study highlights the process of transparently developing a hierarchy of chemical schemes traceable to a benchmark mechanism that reflects the state-of-the-science in chemical understanding (Kaduela et al., 2015, doi:10.1016/j. atmosenv.2015.10.031).

The authors demonstrate a directed graph path-based automated model reduction approach, going through the necessary steps needed to reduce complex atmospheric chemical mechanisms, such as that for isoprene degradation, including optimisation and evaluation. This approach is certainly one of the main ways forward that atmospheric chemists should be using for dynamically constructing chemical mechanisms for a range of applications.

**Comment 1:**

The motivation and application of this study are well founded and reasonably well executed. However, I am not clear on what the main focus of this work is. Is it to demonstrate the first steps in using graph theory in the development of reduced chemical mechanisms from benchmark descriptions of (atmospheric) chemistry, or is it the development of a new reduced isoprene scheme, optimised and evaluated for

specific conditions for use in US regulatory models? It could be both, but the paper should then be split into two clear sections, focused on these two motivations.

For example, the graph theory method sections are really interesting and initially set the paper off as a description of a new method for reducing complex chemical mechanisms. This is great and it's clear that a lot of effort has gone into the approach. However, this is largely undone by the requirement for the manual steps outlined in sections 2.3.5 and 2.4. The changes made here are quite substantial (adding a handful of reactions to a mechanism with 22 reactions soon represents a high proportion of the total mechanism) and so this paper does not demonstrate a method for automated mechanism reduction but rather is aiming to show off a new isoprene mechanism, optimised for a certain specific range of conditions.

We are grateful for the reviewer's overall positive assessment of this work. Indeed, the centerpiece of the paper is the new reduced isoprene mechanism rather than the reduction method, as was reflected, for example, in the title of the manuscript. That said, we note that graph theory motivated the design of our algorithm, and the concepts we used carried over to our manual optimization steps as well, where giving each mechanistic pathway (sequences of connected species in the graph) unique outputs was crucial in optimizing the mechanism. As mentioned in the final paragraph, a more fully automated model reduction method, building from lessons learned in this study, is currently under development and will be presented in a subsequent paper. We have gone through the manuscript to further clarify and emphasize these points in the revised text.

Line 19: "This work demonstrates a new highly reduced isoprene mechanism and shows the potential value of automated model reduction for complex reaction systems."

After line 85: "The AMORE-Isoprene mechanism was the product of this methodology. Our novel algorithm was essential in the creation of this mechanism, but requires further work before it can be used for other mechanisms and without manual adjustment."

After line 308 of revision (Section 2.4): "The graph theoretical framework helped inform our decisions in this process. For example, the conceptualization of the mechanism as a set of unique pathways connected by sequences of reactions, which is rooted in graph theory, helped us to categorize reactions and how adjustments to their parameters would impact end results under different testing conditions."

Also see the concluding paragraph (unchanged) of the paper (Lines 602-605):

*"The AMORE-Isoprene mechanism demonstrates that there is significant potential advantage in the use of algorithms for model reduction. Additional development,*

informed by the experiences of this study, is underway to more fully automate the model reduction process and further reduce the need for manual adjustments. Future work will extend this work to application to reduction of a wide range of atmospheric chemical mechanisms in addition to the isoprene oxidation mechanism."

**Comment 2:**

With this in mind, it feels like the validation/evaluation of the mechanism in section 3 focuses quite heavily on comparison to other mechanisms rather than comparison to measured data (whether from limited chamber experiments or ambient measurements). Again, if this were a paper displaying a mechanism reduction technique then it would be reasonable to make comparisons only to the mechanism you have reduced, but since there have been extensive manual changes to form the mechanism, this seems like more of an exercise to produce a good isoprene mechanism, in which case comparison to real-world data is required.

We agree that comparison to experimental and field data is important for model validation. This is why the model results were compared to chamber data in the context of the box model testing (e.g., section 3.2 and Figure 7) and to ambient data after implementation into CMAQ (section 3.3 and Figure 8). In the interest of brevity, not all testing results were shown in the original manuscript. To expand on what was shared in the original manuscript, additional binned bias plots similar to those in Figure 8 (in the original manuscript, now Figure 9 in the revision) for NOy, OC, and Isoprene have been added to the supplement Section 16.

This added Supplement section is shown here for ease of reference:

"Additional CMAQ results are shown in figure S.12. These plots show the binned bias of the AMORE mechanism for different ranges of the measured value. All measurements are part of the LISTOS campaign. The AMORE mechanism (red) is shown in comparison to the CRACMM1 base mechanism (gray) and a prior version of the AMORE mechanism (blue).

The AMORE mechanism shows slightly very slightly increased OC concentrations and slightly decreased isoprene concentrations, however the difference is not significant. There is no discernible difference in the NOy concentrations between the three mechanisms. Due to issues with the CMAQ run boundary conditions, the overall magnitude of the bias is confounded by many factors unrelated to isoprene. Thus, the

**main conclusion to be drawn from these graphs is that the overall changes for these species is low."**

**AMORE mechanism evaluation: Isoprene biases**

AMORE mechanism evaluation: NOy biases

Based on the reviewer comments, we also concluded that more validation of the expanded Caltech mechanism we used as a basis for the model reduction would be helpful to put the paper and mechanism intercomparisons on a stronger footing. For more information on this please see the response to Comment 6, below.

**Comment 3:**

I also do not think that it has been demonstrated how well this mechanism performs under different atmospheric conditions for applications in other regional models. The input parameters used in the pathway importance algorithm and the model scenarios outlined in Table 2 and Table 3 do not demonstrate an ability to work in high-NOx environments. The High-NOx case included is 5 ppbv. How would this mechanism work, for example, in urban conditions in China (where ozone titration of NO effects the NO/NO2 ratio as the day progresses) or even modelling atmospheric chemistry over the Bornean rain forest?

Based on the comments of both reviewers, we have investigated an expanded range of conditions (including elevated values for NO, NO2, and OH) both for assessing the performance of AMORE Isoprene 1.0 and for assessing the sensitivity of the AMORE algorithm. Performance of AMORE Isoprene 1.0 is comparable or better than other small isoprene mechanisms, giving us confidence that this mechanism is suitable for low and high NOx (and high ozone) scenarios. The added supplement section, S.15, is given here for reference and further discussion:

**"S.15. Additional AMORE Box Model Simulations**

Only six testing conditions were shown in the box model testing for the AMORE-Isoprene mechanism. Here we include additional box model results at more extreme conditions that are not well represented in the six main testing conditions. From these additional tests, we conclude that the AMORE-Isoprene mechanism is suitable for these additional extreme conditions.

The first additional testing condition was with very low  $NO_x$  concentrations. The initial  $NO_2$  concentration was set to 0.05 ppb. The rest of the testing inputs are shown in the error table provided below. Plots of  $NO_2$ , OH, and ozone are provided as well. At low  $NO_x$ , the AMORE-Isoprene mechanism continues to have low error for OH,  $NO_x$ ,  $O_3$ , and other important species. As demonstrated by the Figure S.9, the AMORE is the most accurate small mechanism in these conditions.

Table S.7 Very low NOx error table

|       |       | ,, -   |             |       |  |
|-------|-------|--------|-------------|-------|--|
|       | AMORE | CRACMM | Caltech Rec | CB6r3 |  |
| IEPOX | 0.13  | 0.58   | 0.10        | 0.22  |  |
| NO    | 0.23  | 0.97   | 0.09        | 0.63  |  |
| NO2   | 0.20  | 0.87   | 0.07        | 0.60  |  |
| НО    | 0.13  | 0.41   | 0.05        | 0.23  |  |
| HO2   | 0.18  | 0.94   | 0.04        | 0.25  |  |
| NO3   | 0.32  | 0.99   | 0.12        | 0.57  |  |
| ISOP  | 0.33  | 0.29   | 0.07        | 0.19  |  |
| 03    | 0.14  | 0.80   | 0.03        | 0.57  |  |
| MO2   | 0.70  | 0.69   | 0.29        | 0.63  |  |
| ACO3  | 0.84  | 0.76   | 0.31        | 0.46  |  |
| PAN   | 0.81  | 0.99   | 0.25        | 0.46  |  |
| HCHO  | 0.45  | 0.86   | 0.24        | 0.35  |  |
| ISOPN | 0.48  | 0.67   | 0.13        | 0.91  |  |
| GLY   | 0.85  | 0.99   | 0.40        | 0.47  |  |
| MGLY  | 0.81  | 0.97   | 0.16        | 0.42  |  |
|       | 6.62  | 11.77  | 2.34        | 6.95  |  |
|       |       |        |             |       |  |

ISOP = 10ppb, photolysis rate = 1, NO2 = 0.05ppb, H2O2 = 200 ppb

Figure S.9 Very low NOx (0.05 ppb) simulation, NO2, OH, and O3 Plots

---

## Author Response (AR2)

Review 2nd Round Response:

Comment: Additional text and figures should be spell checked and it made sure that the new supplementary material is properly linked to the associated sections in the main text.

*Response: All references to the SI and new figures have been checked, and fixes made where needed. All changes are shown in the tracked change document.*

Comment: the EUROCHAMP and FIXCIT chamber experimental data and databases need to be properly referenced.

*Response: The EUROCHAMP database campaign is cited in the main text. The FIXCIT database is not mentioned in the main text, it is mentioned along with EUROCHAMP in the Supporting Information. Both are now cited in the references section of the supporting information, which has been added to the end of the document. For reference, here is the references section of the Supporting Information document:*

*References:*

1. *Bates, K. H. and Jacob, D. J.: A new model mechanism for atmospheric oxidation of isoprene: global effects on oxidants, nitrogen oxides, organic products, and secondary organic aerosol, Atmospheric Chemistry and Physics, 19, 9613–9640, https://doi.org/10.5194/acp-19-9613-2019, 2019.*
2. *Muñoz, A.: Isoprene+NO+hydrogen peroxide + OH - Gas-phase oxidation - product study, Tech. rep., CEAM, url https://data.eurochamp.org/data-access/chamber-experiments/25320384-8599-4bc9-a35d-993a77cec7c, 2021a.*
3. *Muñoz, A.: Isoprene+ozone+carbon monoxide + O3 - Gas-phase oxidation - product study, Tech. rep., CEAM, url https://data.eurochamp.org/data-access/chamber-experiments/25320384-8599-4bc9-a35d-993a77cec7c, 2021b.*
4. *Muñoz, A. and Gómez-Alvarez, E.: Overview Of The Eurochamp Database Of European Atmosphere Simulation Chambers, in: Simulation and Assessment of Chemical Processes in a Multiphase Environment, pp. 61–70, Springer, 2008.*
5. *Nguyen, T. B., Crounse, J. D., Schwantes, R. H., Teng, A. P., Bates, K. H., Zhang, X., St. Clair, J. M., Brune, W. H., Tyndall, G. S., Keutsch, F. N., Seinfeld, J. H., and Wennberg, P. O.: Overview of the Focused Isoprene eXperiment at the California Institute of Technology (FIXCIT): mechanistic chamber studies on the oxidation of biogenic compounds, Atmospheric Chemistry and Physics, 14, 13 531–13 549, https://doi.org/10.5194/acp-14-13531-2014, 2014.*
6. *Vasquez, K. T., Crounse, J. D., Schulze, B. C., Bates, K. H., Teng, A. P., Xu, L., Allen, H. M., and Wennberg, P. O.: Rapid hydrolysis of tertiary isoprene nitrate efficiently removes $NO_x$ from the atmosphere, Proceedings of the National Academy of Sciences, 117, 33 011–33 016, https://doi.org/10.1073/pnas.2017442117, 2020.*

7. *Wennberg, P. O., Bates, K. H., Crounse, J. D., Dodson, L. G., McVay, R. C., Mertens, L. A., Nguyen, T. B., Praske, E., Schwantes, R. H., Smarte, M. D., St Clair, J. M., Teng, A. P., Zhang, X., and Seinfeld, J. H.: Gas-Phase Reactions of Isoprene and Its Major Oxidation Products, Chemical Reviews, 118, 3337–3390, https://doi.org/10.1021/acs.chemrev.7b00439, pMID: 29522327, 2018.*